# Microbiome research in general and business newspapers: How many microbiome articles are published and which study designs make the news the most?

**Andreu Prados-Bo** [1,2] *, **Gonzalo Casino** [1,3] *

**1** Department of Communication, Pompeu Fabra University, Barcelona, Spain, **2** Blanquerna School of Health Sciences, Ramon Llull University, Barcelona, Spain, **3** Iberoamerican Cochrane Centre, Biomedical Research Institute Sant Pau (IIB Sant Pau), Barcelona, Spain

* andreu@andreuprados.com (APB); gonzalo.casino@upf.edu (GC)

**Data Availability Statement:** All relevant data are within the paper and its Supporting Information files.

## Abstract

The microbiome is a matter of interest for science, consumers and business. Our objective is to quantify that interest in academic journals and newspapers, both quantitatively and by study design. We calculated the number of articles on the microbiome from the total number of biomedicine articles featured in both PubMed and Spanish science news agency SINC, from 2008 to 2018. We used the Factiva database to identify news stories on microbiome papers in three general newspapers (*The New York Times*, *The Times* and *El País*) and three business newspapers (*The Wall Street Journal*, the *Financial Times* and *Expansión*), from 2007 to 2019. Then, we compared news stories with microbiome papers in PubMed, while also analyzing the frequencies of five study design types, both in the newspapers and in the papers themselves. Microbiome papers represented 0.8% of biomedicine papers in PubMed from 2008 to 2018 (increasing from 0.4% to 1.4%), while microbiome news published by SINC represented 1.6% of total biomedical news stories during the same period (increasing from 0.2% to 2.2%). The number of news stories on microbiome papers correlated with the number of microbiome papers (0.91, p < 0.001) featured in general newspapers, but not in business ones. News stories on microbiome papers represented 78.9% and 42.7% of all microbiome articles in general and business newspapers, respectively. Both media outlet types tended to over-report observational studies in humans while under-reporting environmental studies, while the representation of systematic reviews of randomized controlled trials, randomized controlled trials and animal/laboratory studies was similar when comparing newspapers and PubMed. The microbiome is receiving increasing attention in academic journals and newspapers. News stories on the microbiome in general and business newspapers are mostly based on research findings and are more interested in observational studies in humans and less in environmental studies compared to PubMed.

**Funding:** The authors received no specific funding for this work.

**Competing interests:** have read the journal's policy and the authors of this manuscript have the following competing interests: APB is a paid consultant to companies commercially involved in the gut microbiota and probiotics. GC has declared no competing interests. This does not alter our adherence to PLOS ONE policies on sharing data and materials.

## Introduction

On 19 December 2007, four years after the completion of the Human Genome Project, the Human Microbiome Project (HMP), conceived as a "second human genome project" [1], was launched. HMP focuses on microbial communities and their genomes on and in the human body, collectively known as the microbiome [2]. Research into the microbiome dates back to the early 20th century [3]. The 21st century, however, has witnessed a paradigm shift regarding the crucial role microbes play in the way ecosystems—from the ocean to the human body—function, rather than only being seen as infectious pathogens [4].

Relationships between microorganisms living in our bodies—especially in the gut—and health and risk of disease are currently a major focus of research, public interest and potential business for the pharmaceutical and health industries [4]. Much research in the field has focused on the link between the microbiome and physical and mental well-being, with current unknowns highlighting the need for advancement [5]. An altered human microbiome has been associated with the development of a wide range of diseases, including inflammatory bowel disease, metabolic syndrome, autoimmune disorders, and brain diseases, although causation has yet to be established [6]. Furthermore, microbiome research also has implications for food production [7] and for achieving an environmentally sustainable future [8].

As scholarship and investment in microbiome research develop, it is also important to address the impact of that research in general and business newspapers, which are relevant sources of information and can influence the decisions of the public, investors, health decision makers and healthcare practitioners [9,10]. Scientific articles mentioned in the lay press receive, on average, more citations in academic journals than comparable publications from the same journal that did not appear in the lay press [11–16]. The factors associated with greater levels of newspaper coverage for scientific papers have also been studied. They include the prestige of the journal [17], the availability of press releases [18–22], the domestic preference of newspapers for journals from their own country [23], and the newsworthiness of the topic [24].

Although nowadays many people do not obtain their information directly from newspapers, news from traditional media outlets continues to dominate the information repertoire of mobile internet users [25,26]. Social media, which has become an emerging source for keeping up with scientific issues [27], also relies heavily on newspapers to disseminate news among young people and adults [28,29].

Despite the current scientific interest in the microbiome, its social impact in newspapers has not been properly analyzed. One relevant way of studying the subject is by analyzing the number of newspaper articles in which authors, papers or journals are cited (called "press citations") [18,23,30]. As such, our first objective was to analyze the extent to which the predictable increase in the number of microbiome papers in recent years has had a parallel impact in the press, in both general and business newspapers, given the potential for harnessing the human microbiome to prevent, diagnose or cure disease.

Previous studies showed that study designs based on weak methodology (i.e. observational and animal or laboratory studies) are more likely to be covered in newspapers than those of superior quality (i.e. randomized controlled trials) [18,31,32]. Microbiome research over the past two decades has mainly focused on characterizing microbiome composition across cohorts of clinical patients and matched controls, and on using animal models to understand the causal mechanisms [33]. We hypothesize that studies' methodological rigor is not a major driver when selecting news stories on the microbiome for coverage by newspapers. Our second objective was to undertake a controlled comparison of the study designs of the microbiome papers featured in newspapers against those of the microbiome papers that appear in PubMed.

Such an analysis allows us to build a picture of microbiome science's current level of maturity, which shapes both society's perception and the decisions individuals make in relation to their health.

## Methods

### Newspaper coverage of microbiome research

Based on previous studies of press coverage of biomedical research [34–37], we used the Factiva database to search for news stories on the microbiome in three general newspapers (*The New York Times*, *The Times* and *El País*) and three business newspapers (*The Wall Street Journal*, the *Financial Times* and *Expansión*) from the United States, the United Kingdom and Spain, respectively. Those three countries were selected as they were representative of three previously identified national patterns (American, British and Western World) of biomedical reporting in the press [23]. Print and online editions of each newspaper were analyzed together, after ruling out duplicate news stories. The period analyzed begins in 2007—when the HMP was launched [38]—and ends in 2019.

Fig 1 shows how news stories on the microbiome were selected and categorized. First, we identified news stories in the Factiva database from the 6 newspapers that mentioned the microbiome and its hyponyms in any part of the text, no matter how many times. We excluded duplicates, infographics and other non-relevant news stories, as well as those in which the term microbiota referred to *Microbiota decussata*, commonly known as Siberian cypress, or plant flora. After filtering out opinion articles and editorials [19], we went on to study the newspapers' interest in the microbiome (first objective), for which we identified news stories that devoted 50% or more of the text length (estimated by word count) to reporting on the microbiome (n = 518) [39]. Then, to analyze which study designs made the news compared to PubMed (second objective), we focused on microbiome news stories that cited at least one

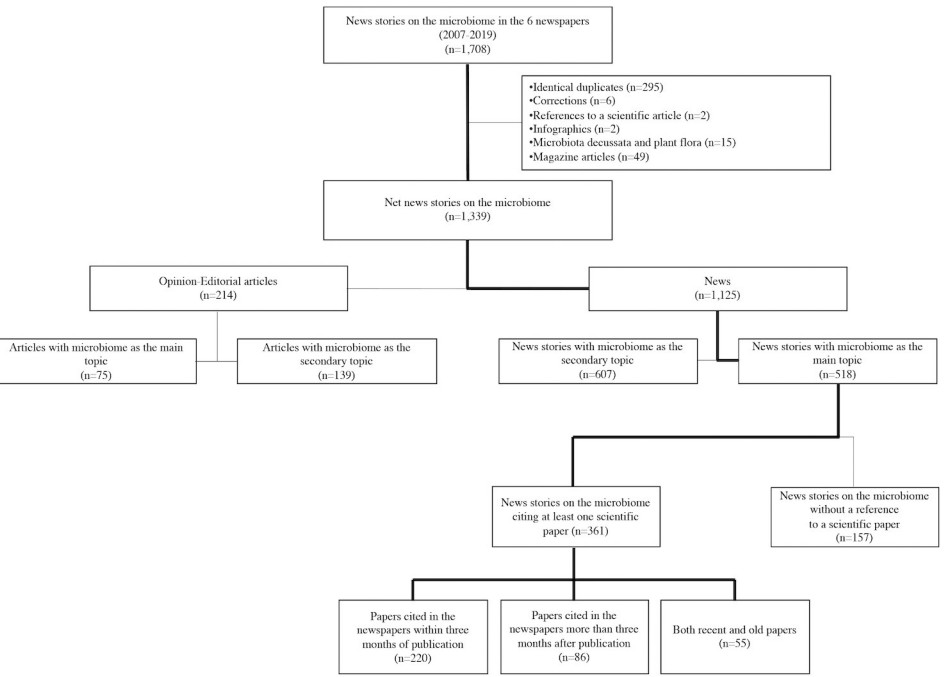

**Fig 1. Process flow diagram of the categorization of news stories on microbiome papers.**

scientific paper (n = 361). For this objective, news stories with the following characteristics were excluded [19]: 1) the microbiome was reported but without reference to a specific study; 2) the microbiome was not a variable of the scientific paper cited in the news story; or 3) the news story was based on premature microbiome research not published in peer-review journals, such as studies presented at scientific or press meetings and ongoing clinical trials registered in the ClinicalTrials.gov database.

Additionally, we studied news interest in the microbiome in the context of biomedicine by quantifying news stories citing the microbiome in the headline and published by SINC, which is a Spanish publicly-funded news agency specializing in science and technology. SINC agency (https://www.agenciasinc.es/) was chosen because it publishes under a Creative Commons 4.0 license and most Spanish newspapers and scientific online sources usually pick up SINC articles to inform the public.

## The microbiome in the context of biomedicine

To estimate the relative interest in microbiome research in the context of biomedical research as a whole, we counted both papers on the microbiome and the total number of scientific publications in the PubMed database of biomedical literature from 2008 to 2018. News agency SINC, which classified news about biomedicine separately until 2018, was used to count the number of news stories on the microbiome against the whole of the biomedicine category. We used data from SINC because the number of news stories on biomedicine in newspapers cannot be calculated using Factiva. Thus, we had an estimate of the press interest in the microbiome from 2008 (when SINC was founded) to 2018 (when it finished categorizing news stories) to compare with the estimate in PubMed.

## Study designs of microbiome papers in PubMed and research news

We adapted the criteria used by Bartlett et al. [18] and Lai and Lane [31] for categorizing the study design of medical research news to microbiome research. As such, we classified the microbiome paper study designs available in PubMed and reported in newspapers into 6 categories: 1) systematic reviews (SRs) with or without meta-analyses of randomized controlled trials (RCTs) in humans; 2) RCTs in humans; 3) human observational studies (defined as prospective and retrospective cohort studies, ecological studies, case-control studies, SRs not of RCTs, and case series); 4) environmental & plant studies (agricultural, aquatic, atmospheric, built environment and terrestrial ecosystems) [40]; 5) animal or laboratory studies; and 6) other designs (interventional studies without randomization and/or without a control group, case reports, narrative or nonsystematic reviews, consensus and reports of expert committees). We excluded commentaries, editorials, perspectives and letters, as such articles do not usually contain research evidence and are not always peer-reviewed [31]. Each scientific study cited in the press was identified on PubMed and downloaded for study design characterization. In PubMed, we set the MeSH and natural terms search to title, abstract and keywords, and applied filters for study designs. It should be acknowledged that PubMed counts articles published in online and print versions separately [41]. The searches were performed by one author (APB) between January and March 2020. Search phrases and filters used in Factiva and PubMed are listed in the supplementary S1 and S2 Files and the data acquired and used for analyses are included in S1 Dataset.

## Statistical analyses

The primary outcome variable was the number of news stories on the microbiome collected by year from 2007 to 2019 that cited at least one scientific paper. That variable was presented as

the mean and standard deviation for the overall sample and subinterest groups: individual newspapers, country (United States, United Kingdom and Spain) and newspaper type (general and business).

The relationship between two quantitative variables was evaluated using the Pearson correlation (linear adjustment): the number of news stories on microbiome papers per year in newspapers and published by SINC and the number of microbiome papers per year in PubMed. The average annual percentage change was evaluated for both the overall sample and the subinterest groups: individual newspapers, country and press type. Finally, the comparison between microbiome news/biomedicine news published by SINC vs microbiome papers/biomedicine papers in PubMed was carried out using a Chi Square test.

The level of significance was set at 0.05. Version 3.5.2 of software R (SPSS Inc., Chicago, IL, USA) and version 4.7.0.0 of the Joinpoint Regression Program were used for all analysis work.

## Results

### Newspaper coverage of microbiome research

Overall, 518 news stories with the microbiome as the main topic were published from 2007 to 2019, of which 361 cited at least one journal article (286 in general newspapers and 75 in business newspapers) (Fig 1). News stories on microbiome papers showed an irregular pattern of evolution compared to microbiome papers in PubMed (Fig 2A). Apart from a peak in 2008, the interest of general newspapers in microbiome research picked up steadily after 2012 with peaks in 2013, 2016 and 2018 (Fig 2B). News stories on microbiome papers represented 77.9% of overall news stories on the microbiome in *The New York Times*, 74.1% in *The Times*, and

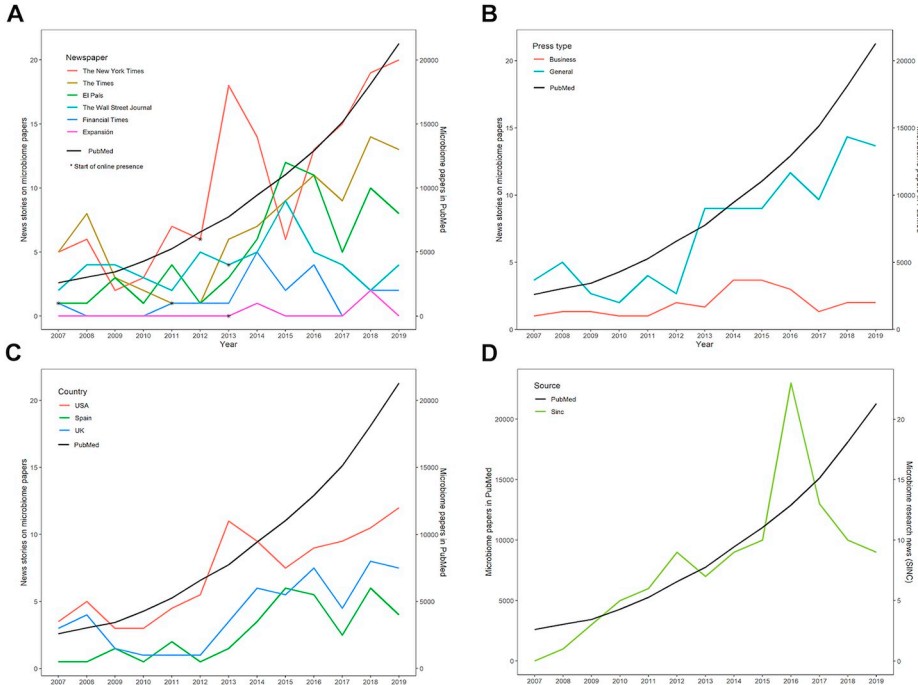

**Fig 2. News stories on microbiome papers compared to microbiome papers in PubMed.** (A) Individual newspapers; (B) General vs business newspapers; (C) Newspapers grouped by countries (the USA, the UK and Spain); (D) Microbiome news/biomedicine news published by SINC (2008–2018) vs microbiome papers/biomedicine papers in PubMed (2007–2019). Microbiome papers in PubMed are presented as a black curve on each graph.

**Table 1. Number of news stories on microbiome papers, microbiome papers in PubMed, microbiome news published by SINC and correlations between them.**

| | Annual cites from 2007 to 2019 | Cites in 2007 | Cites in 2019 | Average annual percentage change | Correlations with microbiome papers in PubMed[1] (p-value) | Correlations with microbiome news published by SINC[1,2] (p-value) |
|---|---|---|---|---|---|---|
| **Microbiome papers in PubMed** | 9297,0 (6063.3) | 2600 | 21292 | 19.6% | - | **0.62 (0.023)** |
| **Biomedicine papers in PubMed** | 1111673,6 (203280.1) | 785933 | 1397557 | 4.9% | - | - |
| **Microbiome/ biomedicine in PubMed** | 0.8% | 0.4% | 1.4% | 9.6% | - | - |
| **Microbiome news in SINC[2]** | 8,1 (5.9) | 0 | 9 | 24.8% | **0.62 (0.023)** | - |
| **Biomedicine news in SINC[2]** | 582.1 (81.1) | 666 | 447 | -3.7% | - | - |
| **Microbiome/ biomedicine in SINC[2]** | 1.6% | 0.2% | 2.2% | 19.5% | - | - |
| **Total newspapers** | 4.6 (4.9) | 2.3 (2.2) | 7.8 (7.5) | 13.9% | **0.88 (<0.001)** | **0.66 (0.014)** |
| **Individual newspapers** | | | | | | |
| *The New York Times* | 10.3 (6.4) | 5 | 20 | 16.0% | **0.83 (0.005)** | 0.48 (0.095) |
| *The Times* | 6.8 (4.4) | 5 | 13 | 14.3% | **0.82 (0.005)** | 0.47 (0.102) |
| *El País* | 5.1 (4.0) | 1 | 8 | 22.7% | **0.74 (0.004)** | **0.71 (0.006)** |
| *The Wall Street Journal* | 4.1 (1.8) | 2 | 4 | 2.9% | 0.14 (0.652) | 0.35 (0.236) |
| *Financial Times* | 1.5 (1.6) | 1 | 2 | 11.8% | 0.39 (0.177) | **0.58 (0.038)** |
| *Expansión* | 0.2 (0.6) | 0 | 0 | 4.3% | 0.41 (0.166) | 0.11 (0.713) |
| **Country** | | | | | | |
| USA | 7.2 (5.6) | 3.5 (2.1) | 12.0 (11.3) | 12.0% | **0.85 (0.002)** | **0.57 (0.039)** |
| UK | 4.1 (4.2) | 3.0 (2.8) | 7.5 (7.8) | 14.5% | **0.81 (0.001)** | **0.57 (0.042)** |
| Spain | 2.7 (3.7) | 0.5 (0.7) | 4.0 (5.7) | 23.1% | **0.75 (0.003)** | **0.68 (0.010)** |
| **Newspaper type** | | | | | | |
| General newspaper | 7.4 (5.4) | 3.7 (2.3) | 13.7 (6.0) | 15.7% | **0.91 (<0.001)** | **0.61 (0.024)** |
| Business newspaper | 1.9 (2.1) | 1.0 (1.0) | 2.0 (2.0) | 7.2% | 0.39 (0.185) | **0.56 (0.043)** |

Mean followed by the standard deviation in parentheses is indicated for microbiome/biomedicine papers in PubMed, microbiome/biomedicine news in SINC and news stories on microbiome papers in newspapers.

[1]The numbers showed the Pearson correlation coefficient.

[2]News stories published by SINC were available from 2008 to 2018.

Significant p-values are highlighted in bold.

78.6% in *El País*. *The New York Times* showed the most intense microbiome research coverage (10.3 news stories on microbiome papers annually), followed by *The Times* (6.8 news stories on microbiome papers annually) and, lastly, *El País* (5.1 news stories on microbiome papers annually) (Table 1).

In business newspapers, news stories on microbiome papers represented 56.4% of the overall number of microbiome news stories for *The Wall Street Journal*, 52.8% for the *Financial Times*, and 18.8% for *Expansión*. *The Wall Street Journal* was the business newspaper that featured microbiome research the most, followed by the *Financial Times* (4.1 and 1.5 news stories on microbiome papers annually, respectively). In contrast, news stories on microbiome papers in *Expansión* were almost null (Table 1).

The strong presence of research in news stories about the microbiome in the press was also supported by significant correlations between the number of news stories on microbiome

papers in general newspapers and the number of microbiome papers in PubMed ($r$ = 0.91, p < 0.001). The magnitude of that association was greater for the American and British general newspapers ($r$ = 0.85, p = 0.002 and $r$ = 0.81, p = 0.001, respectively) than for their Spanish counterpart ($r$ = 0.75, p = 0.003) (Table 1). The strong interest in microbiome research shown by American newspapers compared to British and Spanish newspapers is also apparent in Fig 2C.

As shown in the PubMed curve (Fig 2A–2C), scientific interest in the microbiome gradually grew after 2007 and then picked up speed around 2011. The percentage of microbiome papers available in PubMed and microbiome news published by SINC against the total number of biomedicine publications increased significantly year on year, with a positive annual percentage change of 9.6% and 19.5%, respectively (Table 1). Of the total biomedical literature available in PubMed from 2008 to 2018, the number of microbiome papers increased from 0.4% to 1.4%. Of all the health and biomedical news stories published by SINC, articles citing the microbiome in the headline went from 0.2% to 2.2% from 2008 to 2018, thus doubling the trends shown by PubMed (Fig 2D and Table 1). The comparison between microbiome news/ biomedicine news published by SINC vs microbiome papers/biomedicine papers in PubMed almost reached statistical significance (p = 0.052).

## Papers covered in the news and number of press citations

In the 361 news stories on microbiome papers, 700 different papers published in scientific journals were cited. Each of those 700 papers was covered in at least one news story in one of the six newspapers. Some papers were covered several times in different news stories from different newspapers or from the same newspaper, resulting in a total of 825 press citations of the 700 papers.

The most cited papers (8 press citations) were "Artificial sweeteners induce glucose intolerance by altering the gut microbiota" (the only paper that was cited in all six newspapers) [42] and "Gut microbiota from twins discordant for obesity modulate metabolism in mice" [43]. Two papers had 7 press citations; 1 paper had 6 citations; 2 papers had 4 citations; 8 papers had 3 citations; 72 papers had 2 citations; and the remaining 613 papers had 1 press citation. The papers were cited in the newspapers generally within three months of publication (64.0% for general newspapers and 48.6% for business newspapers).

## Study designs of microbiome papers in PubMed and research news

The study design of microbiome papers available in PubMed between 2007 and 2019 were as follows: 1.8% were SRs of RCTs in humans; 10.9% were RCTs in humans; 8.5% were human observational studies; 46.5% were environmental & plant studies; 30.4% were animal or laboratory studies; and 1.9% had other designs (see methods).

Fig 3 illustrates the over-representation (the percentage of microbiome study design in the press was higher than in PubMed) or under-representation (the percentage of microbiome study design in the press was lower than in PubMed) of microbiome study designs in newspapers vs PubMed. A common pattern was observed among the general and business newspapers that was characterized by an over-representation of observational studies in humans and an under-representation of environmental & plant studies. In contrast, SRs of RCTs in humans, RCTs in humans and animal or laboratory studies tended to be represented to the same degree in newspapers as in PubMed (Fig 3A and 3B).

In terms of study design, no major differences were found between countries (Fig 3B). However, particular features were observed when analyzing newspapers at an individual level. *The New York Times* showed an over-representation (five-fold increase) of SRs of RCTs in

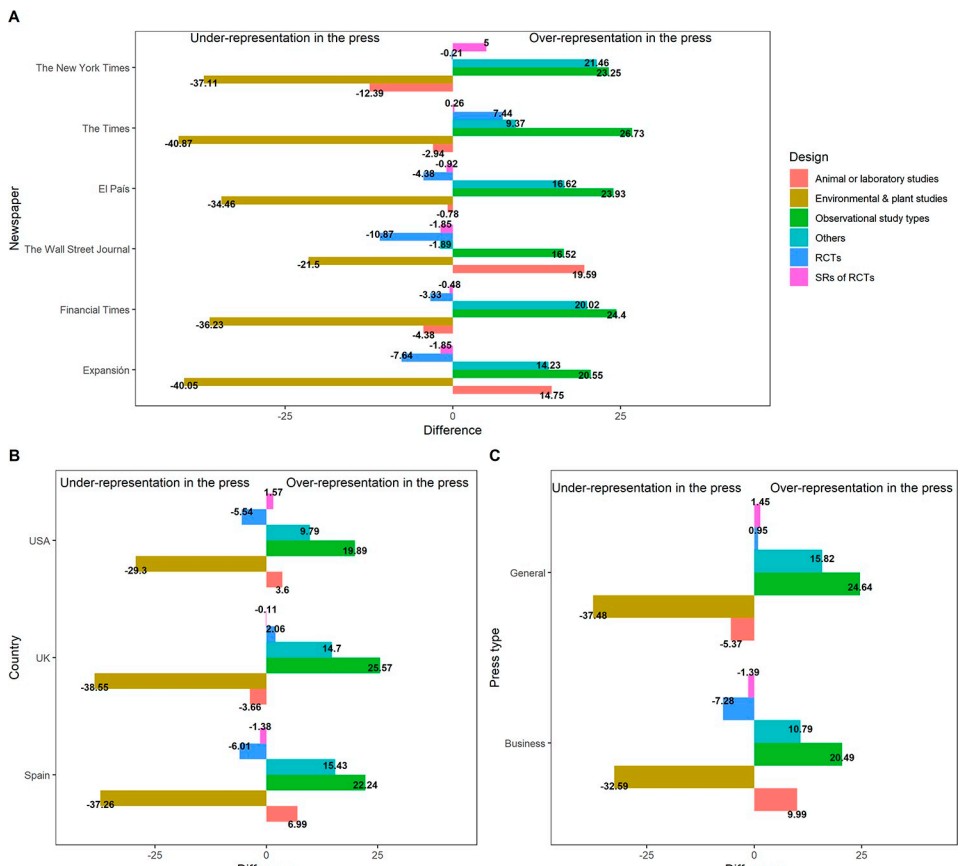

**Fig 3. Over-representation and under-representation of microbiome study designs in the press vs PubMed.**

humans, while *The Times* over-represented (seven-fold increase) RCTs compared to PubMed. In contrast, business newspapers covered a similar number of observational studies in humans and animal/laboratory studies (58/181 and 54/181), with the latter over-represented in *The Wall Street Journal* (Fig 3A).

## Discussion

This is the first study to explore how high-circulation newspapers cover microbiome research in terms of the number of news stories and study design compared to PubMed. Our analysis shows some patterns across newspapers.

First, it should be noted that the proportion of papers on the microbiome in relation to all papers in PubMed has increased steadily from 2007 to 2019, with significant year-to-year growth. That proportion is difficult to estimate for newspapers as they do not classify biomedicine news separately from other news, but a similar increase is shown in publications by SINC, which categorized biomedical news separately until 2018.

Second, an increase in the number of microbiome papers from 2007 to 2019 has resulted in newspapers paying considerable attention to microbiome research, albeit not in a balanced proportion between newspapers. The first peak in press coverage of microbiome research in general newspapers, which took place in 2008, can be explained by the fact that that was the year when large human microbiome research initiatives were launched. They included the first phase of the HMP, the International Human Microbiome Consortium, the EU's

Metagenomics of the Human Intestinal Tract (METAHIT) project and the Canadian Micro-biome Initiative, among others [40]. The subsequent publication of milestone papers in June 2012, reporting on five years of research, triggered an increase in the newspapers' interest in the microbiome [44]. A news peak in 2013 was observed in *The New York Times*, with half of the stories covering research into microbes' role in obesity and cardiovascular disease. That sudden increase also reflects the launch of the second wave of human microbiome projects, including the second phase of the HMP, the European Commission-funded My New Gut pro-gram and the French Government's MetaGenoPolis program [40].

Third, business newspapers are not as sensitive to the two waves of microbiome research projects as general newspapers. That may have two explanations. On the one hand, micro-biome-related patents have increased less markedly than scientific publications [45]. On the other hand, no microbiome therapeutics requiring US Food and Drug Administration and European Food Safety Authority or European Medicines Agency scrutiny have been approved for human use as yet [46].

Fourth, the American and British newspapers were the ones to mention microbiome papers the most. That is not surprising because the US-based National Institutes of Health (NIH) has provided nearly two-thirds of funding for microbiome research [45]. The findings also reflect American and British journalism's longer scientific tradition and the countries' dominant position in the scientific literature [23]. British and American newspapers also echo the top medical journals the most, with *The New York Times* standing out [23]. Similarly, *The New York Times* covered the Human Genome Project the most [1].

Fifth, the abundance of observational studies in humans in newspapers may be rooted in the over-representation of this kind of study design in press releases from journals and institu-tions [18,21], which can influence the content of subsequent microbiome news stories [22]. Our results are in agreement with those of Lai and Lane, who found that English-language gen-eral and business newspapers were more likely to cover observational studies and less likely to feature SRs of RCTs, RCTs and animal/laboratory studies [31]. Similar to our findings in gen-eral newspapers, the authors identified a similar percentage of systematic reviews of RCTs and animal/laboratory studies in the press (3% and 17%, respectively), but did not provide a com-parison group of study designs available in PubMed [31]. Previous research showed that, when choosing observational studies, the press covers study designs of a lower quality (such as those with smaller sample sizes) compared to those published in high impact medical journals. That, in turn, might contribute to distorting the end image of medical advances [32]. The remark-able under-representation of environmental & plant studies in the newspapers under analysis might be rooted in the fact that these studies may be less newsworthy because they do not have a direct impact on human health. Indeed, surveys have shown that the topics of greatest inter-est to society are those of medicine and health, with scientific and technological discoveries and the environment and ecology generating far less interest [9,47]. Second, the level of knowl-edge required for journalists to understand and communicate the findings of these kinds of studies in layman's terms can be higher than that required for observational study types, which could lead to their under-representation in the press. Third, environmental & plant studies are not usually published in top journals, which generally issue press releases, and that can have a negative effect on their impact in newspapers [18–22].

It is also important to acknowledge that the over-representation of observational study designs can distort the public's perceived image of the microbiome, as those study design types are often reported inaccurately in newspapers and usually do not mention any associated cave-ats and limitations [48]. For instance, while an altered microbiome has been reported in a wide variety of health conditions, such as irritable bowel syndrome, obesity and depression, often, it cannot be determined whether differences in the microbiome are causing the disease

or, conversely, if the disease itself is causing the differences [49]. The fact that SRs of RCTs, RCTs and animal or laboratory studies tended to be represented to the same degree in newspapers compared to PubMed, with animal or laboratory studies being the second type of study design most cited in PubMed, reflects how microbiome research is still mainly based on basic science. It is worth highlighting that the reporting of SRs of RCTs and RCTs in *The New York Times* and *The Times*, respectively, may be seen as an indicator of good quality journalism. It should also be noted that the USA and the UK are the countries that produce the most Cochrane SRs [50] and that newspapers tend to report more on domestically produced science [23]. On the other hand, the over-representation of animal/laboratory studies in the business newspapers, which is mainly down to *The Wall Street Journal*, is expected as preclinical microbiome research (representing 30.4% of microbiome studies in PubMed) is the first step towards developing microbiome therapies, and that is where most companies' initial efforts begin. Only a handful of microbiome-related products have entered the end phase of clinical trials [46] and that is reflected in reduced coverage of RCTs in the business press.

Beyond the study design of microbiome papers, the fact that we focused on influential newspapers in terms of readership and circulation might explain the intense coverage of scientific articles about the microbiome, as these newspapers usually have large science and medicine sections with specialist reporters [23]. Other factors might also explain why some microbiome papers are finally echoed by newspapers. They include the impact factor of the journal [16,17], the availability of press releases [18–22], the domestic preference of newspapers for journals from their own country [23], and the newsworthiness of the topic [24]. The last factor is especially relevant in the case of the microbiome, due to the ever-increasing interest among both researchers and the lay public in targeting the microbiome to maintain health and quality of life [4]. An overall analysis considering all of these factors is needed to better elucidate how microbiome research is echoed in the media.

In the light of the ever-increasing amount of research about the link between the microbiome and human health and disease [4], one of the field's urgent needs is precisely that of ensuring unbiased communication of microbiome research to the general public. In that regard, some journals published by the BMJ group indicate the evidence type and subjects studied to journalists when sending embargoed press releases, which may help inform reporting on microbiome findings [51,52]. Keeping up with the huge amount of research and publications on the microbiome and receiving training in science communication skills are also necessary for communicating microbiome research with caution and free from misinterpretation [4]. The European Society of Neurogastroenterology & Motility's Gut Microbiota for Health platform (https://www.gutmicrobiotaforhealth.com/) is an example of a project that aims to translate the latest research on the rapidly-evolving field of the gut microbiome for both the scientific community and the lay public.

Although scientific interest in microbiome research has driven an increase in news stories based on research findings, the patterns observed in study design coverage need to be tracked in order to inform on the evolution of the science behind the current microbial momentum being experienced by society. For microbiome scientists, the coverage of their research in newspapers and its dissemination in social media can improve their visibility and scientific citations. It is also important to acknowledge the potential role of media coverage in obtaining research funding, without forgetting that, although mentions of scholarly outputs on social media and news sites are becoming increasingly present in policy papers and research calls, it is too early to consider whether they contribute to the awarding of research funding [53].

One strength of our study is that we focused on both general and business newspapers ranked high in circulation from different countries over a long period of 12 years. Rather than gauging only news stories considered immediately newsworthy (that is, generated in response

to a paper within 2 months of its publication), we analyzed all news stories on microbiome papers regardless of paper publication date. In addition, we analyzed the number of news stories on microbiome papers and study designs reported in newspapers vs patterns in PubMed.

Our study also has limitations. We did not focus on studying the impact of microbiome research in other mass media such as low-circulation newspapers, magazines, radio, television or the Internet. Moreover, our selection of international general and business newspapers is not representative of the general and business press around the world, even though our selection includes some of the most widely read and best quality international newspapers. Furthermore, our analysis only focuses on quantitative aspects. As a result, the study of all representations of the microbiome in the selected newspapers, regardless of whether they cite a scientific study, is limited in scope and deserves the application of a qualitative methodology that is outside the scope of our research objectives. Other authors have previously addressed newspaper coverage of the microbiome based on qualitative aspects, such as the tone of the discourse [54] and language employed to discuss advances in the microbiome [55], highlighting the need for microbiology literacy in society due to the role of microbes in the health of our planet [56]. Finally, analyzing the press citations of authors, papers or journals also has its limitations, given that their mentioning in newspaper articles does not provide any information about the context of the citation or the quality of the journalistic text.

## Conclusions

Our results show that the microbiome is receiving increasing attention in both research and the press. News stories on the microbiome in both the general and business press during the period under study were mostly based on research findings. While the volume of microbiome-based scientific studies in the press mirrors the number of scientific papers in PubMed, the choice of studies covered by general and business newspapers over-represent observational studies and under-represent environmental & plant studies, while showing a similar degree of representation for SRs of RCTs, RCTs and animal or laboratory studies.

## Supporting information

**S1 File. Factiva search filters and phrases.**
(DOC)

**S2 File. PubMed search filters and phrases.**
(DOC)

**S1 Dataset. Data acquired and used for analyses.**
(XLSX)

## Acknowledgments

We wish to thank Mireia Bosch, Juan Carlos Martín and Queralt Miró for their support with data management and statistical analyses, and María García-Puente, Alicia Jarillo and Marta Diaz for their assistance with searches in PubMed. We also appreciate the critical insights received from Prof. Paul Enck while preparing the manuscript.

## Author Contributions

**Conceptualization:** Andreu Prados-Bo, Gonzalo Casino.

**Data curation:** Andreu Prados-Bo.

**Formal analysis:** Andreu Prados-Bo, Gonzalo Casino.

**Investigation:** Andreu Prados-Bo.

**Methodology:** Andreu Prados-Bo, Gonzalo Casino.

**Project administration:** Andreu Prados-Bo.

**Supervision:** Gonzalo Casino.

**Validation:** Andreu Prados-Bo, Gonzalo Casino.

**Visualization:** Andreu Prados-Bo.

**Writing – original draft:** Andreu Prados-Bo.

**Writing – review & editing:** Gonzalo Casino.

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
