## [Decision Letter · Decision Letter 0]

7 Sep 2020

PONE-D-20-23472

Microbiome research in general and business newspapers: which study designs make the news?

PLOS ONE

Dear Dr. Prados-Bo,

Thank you for submitting your manuscript to PLOS ONE. After careful consideration, we feel that it has merit but does not fully meet PLOS ONE’s publication criteria as it currently stands. Therefore, we invite you to submit a revised version of the manuscript that addresses the points raised during the review process.

We look forward to receiving your revised manuscript.

Kind regards,

Bright Nwaru

Academic Editor

PLOS ONE

Additional Editor Comments:

None

Journal Requirements:

"I have read the journal's policy and the authors of this manuscript have the following competing interests:

APB is a paid consultant to companies commercially involved in the gut microbiota and probiotics. GC has declared no competing interests."

Reviewers' comments:

Reviewer's Responses to Questions

**Comments to the Author**

1. Is the manuscript technically sound, and do the data support the conclusions?

Reviewer #1: Partly

Reviewer #2: Partly

2. Has the statistical analysis been performed appropriately and rigorously? 

Reviewer #1: Yes

Reviewer #2: Yes

3. Have the authors made all data underlying the findings in their manuscript fully available?

Reviewer #1: No

Reviewer #2: Yes

4. Is the manuscript presented in an intelligible fashion and written in standard English?

Reviewer #1: Yes

Reviewer #2: Yes

5. Review Comments to the Author

Reviewer #1: In the manuscript entitled “Microbiome research in general and business newspapers: which study designs make the news?" the Authors provide a comprehensive statistical overview of a time span of a decade of microbiome related articles in three general newspapers, three business newspapers and from scientific articles featured in PubMed, as well as in the science news agency SINC. Besides showing the increased relative abundance of microbiome associated articles in these newspapers and in the scientific database PubMed, the authors characterized the design type of the primary literature articles allowing insight onto the impact on the press. It is an interesting analysis, but the scope and conclusions are rather limited. To assure a better quality of the submitted paper, there are several suggestions to be considered:

Major points

1. The exact and long-term aim of such analysis could be better explained. It is important for researchers to translate their results to the general public. What they can improve?

2. Are the news coverages coming from the article press-releases? This is important to address.

3. It would be interesting to analyze if the citations of the paper depend on press coverage.

4. The remarkable under-representation in the press of environmental & plant studies should be better explained in the discussion. A potential reason could be that environmental & plant studies are harder to apprehend in context of the microbiome to make reader-friendly stories in the press. (?)

5. The limitations of the study by not including the impact of microbiome research in other media have been mentioned in the discussion. A short explanation of the methodological limitations using other types of media could underline the reason for the restricted methods with “only” six newspapers analyzed.

Minor points

1. The visualization of filtering process in Fig.1 could be improved by highlighting the selected paths for analysis.

Reviewer #2: In the current manuscript the authors performed an analysis of microbiome study coverage in general and business newspaper from 2008-2018. The topic is potentially of interest for the scientific community to reach the best coverage of their finding in the news. However, there are several limitations due to the reported findings.

1) For this reviewer it is completely unclear how the authors evaluated the media coverage of certain articles. As far as I interpreted the reported data, the authors have classified articles available on PubMed regarding study evidence. I would prefer to have a comparison between top quality RCTs regarding media coverage. What was the reason for reports in the media? This would support scientists in getting their research recognized by lay audience via media coverage.

2) What was the difference between the three countries and did the authors evaluate reasons for the observed differences?

3) There should be a fact box included on game changers regarding media coverage of scientific data.

4) Was the media coverage accompanied by better funding of research groups? This would be essential information that could enhance the impact of the reported data.

5) Additionally the authors only report regarding newspapers, which is not targeting the younger population. What about social media, online publications,..?

6) In the current form, the data of the manuscript might be better reported in a letter to the editor.

Minor comment:

1) Fig.1. Please specify what Op-Ed is.

2) Please provide more information about SINC (country, city) and why this agency was chosen.

6. PLOS authors have the option to publish the peer review history of their article (what does this mean?). If published, this will include your full peer review and any attached files.

Reviewer #1: No

Reviewer #2: No

---

## [Author Response · Author response to Decision Letter 0]

29 Oct 2020

Academic editor’s comments:

Journal requirements

01 Please ensure that your manuscript meets PLOS ONE’s style requirements, including those for file naming. 

Both new files ‘Revised Manuscript with Track Changes’ and ‘Manuscript’ meet PLOS ONE’s style requirements regarding title, author, affiliations and main body. We have also updated file naming for figures and for supporting information, which have been resubmitted to meet PLOS requirements.

02 Thank you for stating the following in the Competing Interests section:

"I have read the journal's policy and the authors of this manuscript have the following competing interests:

APB is a paid consultant to companies commercially involved in the gut microbiota and probiotics. GC has declared no competing interests."

We have included our updated Competing Interests statement in the cover letter so the online submission form may be changed on our behalf.

Reviewers’ comments: review comments to the author

Reviewer #1

Major points

01 The exact and long-term aim of such analysis could be better explained. It is important for researchers to translate their results to the general public. What they can improve? 

We have changed the way we state the aim and scope of our analysis in the introduction and discussion of the manuscript.

02 Are the news coverages coming from the article press-releases? This is important to address. 

Thank you for acknowledging this point. Although this objective has not yet been studied in the context of the microbiome, it has been subject to considerable study for different scientific and biomedical topics (Entwistle, 1995; De Semir et al., 1998; Stryker, 2002; Schwartz et al., 2012). For instance, one of the manuscript’s authors has documented that journalistic articles in the Spanish press discussing current biomedical issues are intensely mediated by press releases (Casino, 2015). 

The study of microbiome-related press releases does not seem to be a priority at the moment and such an exercise will require another methodology that is outside the scope of our manuscript.

We have clarified in the introduction of the manuscript that the influence of press releases on news coverage of scientific and biomedical topics has been widely studied.

03 It would be interesting to analyze if the citations of the paper depend on press coverage. 

Thank you for your observation. Previous studies cited in the manuscript’s introduction have shown that scientific articles that receive press coverage have, on average, more citations in the scientific literature compared to those not mentioned in the press. 

Exploring to what extent citations of microbiome papers depend on press coverage is outside the scope of this manuscript.

In the introduction of the first submitted version of the manuscript, we cited studies (references 11-16) supporting the influence of press coverage on subsequent scientific citations. 

04 The remarkable under-representation in the press of environmental & plant studies should be better explained in the discussion. A potential reason could be that environmental & plant studies are harder to apprehend in context of the microbiome to make reader-friendly stories in the press. (?) 

Thank you for encouraging us to reflect on an explanation for the under-representation in the press of environmental & plant studies.

We have explained in the discussion that the remarkable under-representation of environmental & plant studies in the newspapers under analysis might have its roots in the fact that the level of knowledge required to understand and communicate their findings in layman’s terms is higher than that required for observational study types. In addition, environmental & plant studies are usually published in less prestigious journals that probably do not issue press releases. Third, the studies may be less newsworthy because do not have a direct impact on human health. We have also acknowledged that the over-representation of observational study designs can distort the public’s perceived image of the microbiome, as those study design types are often inaccurately reported in newspapers and usually do not mention any associated caveats and limitations.

05 The limitations of the study by not including the impact of microbiome research in other media have been mentioned in the discussion. A short explanation of the methodological limitations using other types of media could underline the reason for the restricted methods with “only” six newspapers analyzed. 

We are aware that we have not included any other mainstream media, social media or blogs in our analysis. 

In response to your insightful comment, we have stated in the introduction that, although most people go online to search for information about scientific issues, newspaper content continues to dominate the online information repertoire over other media types. 

Analyzing press citations has advantages over alternative metrics (known as altmetrics) for scientific publications in websites, blogs and social media. That is because the press is the main news producer and allows for reproducible tracking of newspaper articles on a specific topic within a defined period of time via the Factiva database, used in previous studies looking at the press coverage of biomedical research. That is not always possible for a content analysis of other types of media. 

In addition, the selection of the sample of six newspapers responds to three patterns of reporting biomedical research in the press—USA, UK and Western World—described by the author Gonzalo Casino (Casino et al., 2017), which has also been acknowledged in the introduction. As such, our sample selection of newspapers might be seen as an indicator of how other newspapers from the same area of the world report microbiome research.

Minor points

01 The visualization of filtering process in Fig.1 could be improved by highlighting the selected paths for analysis. 

We agree with your suggestion for improving the visualization of the methodology used in our manuscript.

We have changed figure 1 of the manuscript and increased the width of the lines to highlight the selected paths that have been used in our data analyses.

Reviewer #2

Major points

01 For this reviewer it is completely unclear how the authors evaluated the media coverage of certain articles. As far as I interpreted the reported data, the authors have classified articles available on PubMed regarding study evidence. I would prefer to have a comparison between top quality RCTs regarding media coverage. What was the reason for reports in the media? This would support scientists in getting their research recognized by lay audience via media coverage. 

In an initial step, we downloaded all news stories on the microbiome (excluding opinion-editorial articles) and read all of them individually to identify the study mentioned, based on the author’s name, the name of the journal or by accessing the provided link to the scientific publication. Then, we proceeded with classifying the study design of the microbiome papers mentioned in the press and used study designs published in PubMed between 2007 and 2019 as a comparison group. 

Analysis of the impact of study designs in the press is an active area of research that has been applied in the context of biomedical news, as stated in the methods section. We have adapted the criteria used by Bartlett et al. and Lai and Lane for categorizing the study design in newspapers in the context of microbiome research (i.e., we have created a new category for environmental & plant studies due to the special relevance of this research in the microbiome field, according to Stulberg et al.). As a result, in our analysis of newspapers and PubMed, we have considered the 6 study design categories defined in our manuscript as follows: SRs of RCTs, RCTs, observational study types, animal or laboratory studies, environmental & plant studies, and other designs. 

In line with the criteria used by Bartlett et al. and Lai and Lane, and given that previous work suggests that large RCTs that report hard outcomes (i.e. high-quality RCTs) usually attract the same press interest as low-quality RCTs, we have not focused solely on the press coverage of high-quality RCTs. However, our classification does differentiate between randomized controlled trials (high-quality RCTs) and intervention studies without randomization and/or without a control group (low-quality RCTs). The latter has been included in the “other designs” category, as stated in the methods section. 

Study quality is evaluated after publication and such an analysis would have required a specific methodology (such as the GRADE system), which is outside the scope of this study. 

In the methods section, we have clarified the method used for classifying all of the different study design types mentioned in microbiome news stories. 

02 What was the difference between the three countries and did the authors evaluate reasons for the observed differences? 

Thank you for acknowledging this point.

First of all, we have clarified in the methods section that the three countries (USA, UK and Spain) were selected as they were representative of three previously identified national patterns of biomedical reporting in the press (American, British and Western World).

We have stated in the results section that American newspapers showed a greater interest in microbiome research, followed by British newspapers and, lastly, Spanish newspapers. When it comes to study design, no major differences were found between countries, as shown in figure 3B. However, at an individual level, The New York Times showed an over-representation of SRs of RCTs in humans, while The Times over-represented RCTs compared to PubMed. 

We have also evaluated in detail in the discussion the reasons behind the observed differences. American and British newspapers’ considerable interest in microbiome research compared to the Spanish press might reflect the USA and the UK’s longer and more far-reaching scientific tradition and that tradition’s dominant position in the scientific literature. Second, British and American newspapers also echo the top medical journals the most. Third, the over-representation of SRs of RCTs and RCTs in The New York Times and The Times, respectively, may be because the USA and the UK are the countries that produce the most Cochrane SRs and newspapers tend to report more on domestically produced science.

03 There should be a fact box included on game changers regarding media coverage of scientific data. 

We agree with the fact that it is important to dedicate special attention in our manuscript to changes in the landscape of coverage of scientific data with the emergence of social media and blogs as news sources.

In the introduction section, we have devoted several paragraphs to discussing online content and social media as new players involved in how scientific data is covered by the media. We also discuss social media’s impact on how the lay public engages with science, but also as a means by which scientists can improve their citations in academic journals. Although the lay public uses the internet as the main source for keeping up-to-date with scientific issues, newspaper content still dominates the online information repertoire and social media are used by newspapers as a way of amplifying their own content.

04 Was the media coverage accompanied by better funding of research groups? This would be essential information that could enhance the impact of the reported data. 

Thank you for your comment on the relationship between the funding of research and media coverage of scientific articles.

There is little data in the literature regarding whether media coverage of scientific articles is used by funding entities to assign money for research and this objective is outside the scope of our manuscript. 

Competitive research funding devotes part of the budget to communication and dissemination activities, which means more visibility in the media for the research group and thus greater probability of obtaining funding. However, media coverage is only one of the factors that can improve the funding of research projects. 

As explained in the discussion, although mentions of scholarly outputs in social media and news sites are increasingly included in policy papers and research calls, it is too early to consider whether they contribute to the awarding of research funding. 

05 Additionally the authors only report regarding newspapers, which is not targeting the younger population. What about social media, online publications,..? 

We are aware that we have not included social media and other online publications in our analysis. 

In the updated version of our manuscript, we have stated in the introduction that even though most people go online to search for information about scientific issues, newspaper content still dominates the online information repertoire over other types of media. In addition, the social media platforms used by the younger population tend to get their information from digital newspaper content. As we are aware of the changing landscape in which traditional media is moving from print to digital platforms, we included in our analysis the digital version of the newspapers under study when they were included in the Factiva database.

06 In the current form, the data of the manuscript might be better reported in a letter to the editor. 

We appreciate the reviewer’s suggestion, but both authors consider that the results of this manuscript are original and novel and therefore are a better fit for the research article category. Furthermore, as far as we know, PLOS ONE does not include letters to the editor.

Minor comments

01 Fig. 1. Please specify what Op-Ed is. 

Fig. 1 has been updated and includes the explanation of what Op-Ed is (i.e., opinion-editorial articles).

02 Please provide more information about SINC (country, city) and why this agency was chosen. 

In the methods section (subsection “Newspaper coverage of microbiome research”), we have provided detailed information about SINC.

---

## [Decision Letter · Decision Letter 1]

22 Feb 2021

PONE-D-20-23472R1

Microbiome research in general and business newspapers: how many microbiome articles are published and which study designs make the news the most?

PLOS ONE

Dear Dr. Prados-Bo,

Thank you for submitting your manuscript to PLOS ONE. After careful consideration, we feel that it has merit but does not fully meet PLOS ONE’s publication criteria as it currently stands. Therefore, we invite you to submit a revised version of the manuscript that addresses the points raised during the review process. In particular, please address the concerns raised by reviewer 3.

We look forward to receiving your revised manuscript.

Sincerely,

Yann Benetreau, Ph.D.

Senior Editor (Staff Editor), *PLOS ONE*

Journal Requirements:

Additional Editor Comments (if provided):

Reviewers' comments:

Reviewer's Responses to Questions

**Comments to the Author**

1. If the authors have adequately addressed your comments raised in a previous round of review and you feel that this manuscript is now acceptable for publication, you may indicate that here to bypass the “Comments to the Author” section, enter your conflict of interest statement in the “Confidential to Editor” section, and submit your "Accept" recommendation.

Reviewer #2: All comments have been addressed

Reviewer #3: (No Response)

2. Is the manuscript technically sound, and do the data support the conclusions?

Reviewer #2: Yes

Reviewer #3: Partly

3. Has the statistical analysis been performed appropriately and rigorously? 

Reviewer #2: N/A

Reviewer #3: Yes

4. Have the authors made all data underlying the findings in their manuscript fully available?

Reviewer #2: Yes

Reviewer #3: Yes

5. Is the manuscript presented in an intelligible fashion and written in standard English?

Reviewer #2: Yes

Reviewer #3: Yes

6. Review Comments to the Author

Reviewer #2: (No Response)

Reviewer #3: The second version of the paper addresses most of the comments made by the first round reviewers. Anyway, there are still some points that need to be considered before publication. They are the following:

a) applying the exclusion criteria proposed by the Authors, the news related to microbioma without an explicit reference to scientific papers remain out of the analysis. This is not a problem per se, but if the Authors claim to study the "social impact" of the microbioma's research the generic discourse about it in the newspapers is a relevant aspect. As a consequence, it cannot be completely ignored and if the Authors decide to leave this aspect outside the paper, it should be discussed, at least;

b) moreover, taking into account the above cited selection criteria, it is not surprising "the strong presence of research in the news stories about microbiome in the press". This result, indeed, mostly depends on precisely the application of such exclusion criteria;

c) it should be explained why the scientific papers most cited in the news have gained such a position. In other words, why these paper are more newsworthy than others?

d) the Authors listed three reasons for explaining “the remarkable under-representation of environmental & plant studies in the newspapers”. The last is that this kind of “studies may be less newsworthy because do not have a direct impact on human health”, but I wonder whether this can be actually the first and the most relevant.

7. PLOS authors have the option to publish the peer review history of their article (what does this mean?). If published, this will include your full peer review and any attached files.

Reviewer #2: No

Reviewer #3: No

---

## [Author Response · Author response to Decision Letter 1]

11 Mar 2021

Academic editor’s comments

Journal requirements

Question 1: Please review your reference list to ensure that it is complete and correct. If you have cited papers that have been retracted, please include the rationale for doing so in the manuscript text, or remove these references and replace them with relevant current references. Any changes to the reference list should be mentioned in the rebuttal letter that accompanies your revised manuscript. If you need to cite a retracted article, indicate the article’s retracted status in the References list and also include a citation and full reference for the retraction notice. 

Answer 1: The reference list of the manuscript has been reviewed in detail. It does not contain papers that have been retracted nor unavailable and unpublished work and personal communications. 

All the references included in the manuscript as it stands now meet criteria specified in PLoS One submission guidelines (https://journals.plos.org/plosone/s/submission-guidelines#loc-references). 

Reviewers’ comments: review comments to the author

Question 1 from reviewer 3: Applying the exclusion criteria proposed by the Authors, the news related to microbioma without an explicit reference to scientific papers remain out of the analysis. This is not a problem per se, but if the Authors claim to study the "social impact" of the microbioma's research the generic discourse about it in the newspapers is a relevant aspect. As a consequence, it cannot be completely ignored and if the Authors decide to leave this aspect outside the paper, it should be discussed, at least.

Answer question 1 from reviewer 3: 

Thank you for acknowledging this point as this question needs to be discussed in depth.

For our first objective of quantifying the six newspapers’ interest in the microbiome, we focused both on news stories on the microbiome that cite at least one scientific paper (361/518=69.7%) and news stories on the microbiome that do not reference a scientific paper (157/518=30.3%), in relation to the total number of news stories that had the microbiome as the main topic. For the second objective of quantifying which microbiome study design made the news the most, we only focused on news stories that explicitly mentioned a scientific publication on the microbiome because we wanted to quantitatively compare that to PubMed publications.

News stories on the microbiome that did not reference a scientific paper can be addressed according to a qualitative methodology that explores several variables, including the topics addressed and the language used to discuss the latest scientific advances in microbiome research. However, this research objective is outside the scope of our manuscript. It should also be acknowledged that other authors have focused on studying news stories on the microbiome from the perspective of language and by using a qualitative methodology (Chuong KH et al. 2015; Nerlich & Hellsten, 2009; Nerlich, 2017).

We have re-written the first part on newspaper coverage of microbiome research in the methods section to better clarify our workflow and analysis. We are aware that excluding news stories that had the microbiome as the central topic but which did not reference a scientific paper is a limitation of our manuscript and have acknowledged this point in the discussion.

Question 2 from reviewer 3: Moreover, taking into account the above cited selection criteria, it is not surprising "the strong presence of research in the news stories about microbiome in the press". This result, indeed, mostly depends on precisely the application of such exclusion criteria.

Answer question 2 from reviewer 3:

Thank you for your comment. 

The microbiome is a current hot topic for newspaper coverage and, in an initial step, we applied strict selection criteria to ensure that the news stories under analysis covered the microbiome as the central topic, in an in-depth and objective way.

If a news story mentioned the microbiome or any of its synonyms in a quote, for contextualizing a recipe or in the title of a book, we did not consider them to be relevant enough in terms of informing on the microbiome. Therefore, articles that mentioned the microbiome at some point but where the microbiome was not the focus—i.e. it was discussed in less than 50% of the text—were excluded from our analysis. This sampling process is known to eliminate news stories that contain tangential microbiome-related content (i.e., false positives) (Guasch et al., 2019).

In our first objective of quantifying the newspapers’ interest in the microbiome, we found that news stories on the microbiome that do not mention a scientific article represented a minority (30.3%), in relation to the total number of news stories that had the microbiome as the main topic. As we were interested in doing a controlled comparison between the scientific literature and newspapers, in our second objective, we ruled out news stories on the microbiome that did not mention a scientific paper in order to compare the same variable (mention of a scientific paper) in the six newspapers and in articles collected via PubMed.

We have acknowledged in the discussion the limitation of not considering all representations of the microbiome, regardless of whether they cite a scientific study, in the newspapers we analyzed.

Question 3 from reviewer 3: It should be explained why the scientific papers most cited in the news have gained such a position. In other words, why these paper are more newsworthy than others?

Answer question 3 from reviewer 3:

Thank you for your observation.

There is no one factor alone that can explain why some microbiome scientific articles are more newsworthy than others. In our analysis we have shown that study design could influence journalists’ selection of the scientific papers that will be echoed in newspapers. Beyond study design, our decision to focus on influential newspapers in terms of readership and circulation might explain the intense coverage of scientific findings in the press, as these newspapers have large science and medicine sections with specialist reporters. Other factors that might also be important include the impact factor of the journal, the availability of press releases, the domestic preference of newspapers for journals from their own country, and the newsworthiness of the topic.

We have clarified in the discussion which factors, beyond study design, may explain why some microbiome scientific papers are more newsworthy than others (lines 351-362). To date, different partial analyses have been done studying which factors shape how scientific and medical findings are echoed in newspapers, while in future an overall analysis could clarify which factors have the strongest weight in determining which microbiome scientific papers are finally echoed in the media.

Question 4 from reviewer 3: The Authors listed three reasons for explaining “the remarkable under-representation of environmental & plant studies in the newspapers”. The last is that this kind of “studies may be less newsworthy because do not have a direct impact on human health”, but I wonder whether this can be actually the first and the most relevant.

Answer question 4 from reviewer 4:

Thank you for encouraging us to reflect on this explanation for the under-representation of environmental & plant studies in the newspapers under analysis.

We agree on your point of view. Therefore, we have stated in the discussion that the fact that environmental & plant studies do not have a direct impact on human health might be the most relevant underlying factor in our findings and have placed it in first position on the list. Moreover, we have also cited the findings of two large population surveys that have shown that the topics of greatest interest to society are those of medicine and health, with scientific and technological discoveries and the environment and ecology generating far less interest (National Science Board, 2018; Fundación Española para la Ciencia y la Tecnología (FECYT), 2018).

---

## [Editor Report · Decision Letter 2]

26 Mar 2021

Microbiome research in general and business newspapers: how many microbiome articles are published and which study designs make the news the most?

PONE-D-20-23472R2

Dear Dr. Prados-Bo,

We’re pleased to inform you that your manuscript has been judged scientifically suitable for publication and will be formally accepted for publication once it meets all outstanding technical requirements.

Kind regards,

Federico Neresini

Guest Editor

PLOS ONE
---

## [Editor Report · Acceptance letter]

30 Mar 2021

PONE-D-20-23472R2 

Microbiome research in general and business newspapers: how many microbiome articles are published and which study designs make the news the most? 

Dear Dr. Prados-Bo:

I'm pleased to inform you that your manuscript has been deemed suitable for publication in PLOS ONE. Congratulations! Your manuscript is now with our production department. 

Kind regards, 

on behalf of

Dr. Federico Neresini 

Guest Editor

PLOS ONE